# Socioecological Transition in Land and Labour Exploitation in Mallorca: From Slavery to a Low-Wage Workforce, 1229–1576

**Gabriel Jover-Avellà** [1,*], **Antoni Mas-Forners** [2], **Ricard Soto-Company** [3] and **Enric Tello** [3]

1   Economics Department, University of Girona, Montilivi Campus, 17003 Girona, Spain
2   Departament of Historical Sciences and Arts Theory, Faculty of Philosophy and Arts,
    University of Balearic Islands, Palma de Mallorca, Cra. de Valldemossa, km 7.5,
    07122 Palma, Spain; antoni.mas@uib.es or antoin.mas@uib.edu
3   Department of Economic History, Institutions, Policy and World Economy, University of Barcelona,
    Diagonal Avenue 690, 08034 Barcelona, Spain; ricardsoto@ub.edu (R.S.-C.); tello@ub.edu (E.T.)
*   Correspondence: gabriel.jover@udg.edu; Tel.: +34-972-418-223

**Abstract:** The permanence of slave labour until the 16th century was a lasting legacy of the late feudal colonization of the Mallorca Island. Through a large set of probate inventories and accounting books, we have documented the use of a great deal of slaves in farming large noble estates during the 14th and 15th centuries. The defeat of the peasant revolt of 1450–1454 offered to nobles and patricians the opportunity to seize much of the land previously colonized by Mallorcan peasants. This creation of a dispossessed peasantry, combined with new trade demands, led to a transition from slave-powered manorial farms to capitalist olive oil-exporting estates that took advantage of the low-wage workforce reserve. A peculiar feature was the massive use of women's gangs as olive pickers when olive oil became the main cash-crop exported from the 16th century onwards. By linking changes in work and land uses, this study brings to Southern Europe the debate over the driving forces of the emergence of agrarian capitalism.

**Keywords:** feudalism; slavery; agricultural low-wage labour market; women hired as farm daily labourers; land use change; socioecological transition

## 1. Feudal Colonialism, Slavery and Wage Labour in the Mediterranean

The late Middle Ages (1350–1550) has been pinpointed from different historiographical approaches as the period when divergences in European agrarian development originated [1]. There is still much debate, however, on the drivers leading to the different paths farm systems were to take [2–5]. Divergences in regional development occurred in three main dimensions: changes in land ownership rights (capitalist farm versus peasant tenure); in labour markets (wage labour versus serfdom), and in farm management (innovations and yield improvements). The combination of these processes led to agricultural and social structures with different trajectories in terms of improving social welfare and stimulating economic growth [6,7]. The aim of this study is to bring this debate to Southern Europe, focusing on the land and labour changes at the interface of socioeconomic and sociological transitions. Its objective is to link labour market changes driven by the disappearance of slave labour and the emergence of a low-wage labour market with the agricultural transformations that took place in a small feudal enclave bordering the western Mediterranean: the island of Mallorca.

In the Mediterranean regions the development of a dense urban fabric, and the emergence of powerful commercial and manufacturing enclaves during the late Middle Ages [8–10], were based on different agrarian class structures: peasant agriculture, manorial estate ownership and even the

premature development of agrarian capitalism [11]. However, the main differences with Atlantic regions can be noted in two features. First, the agro-ecological factors imposed some restrictions on models of agricultural development, making it impossible to extensively replicate many innovations adopted in Atlantic soil [12]—even though some of them had been previously developed on a small scale in the Mediterranean [13]. Climate conditions and variations were also quite different [14]. Secondly, the Mediterranean continued to be a region of conquests until the 15th century. The feudal frontier continued to be extended, either in a north-south direction as in the Iberian Peninsula, or from east to west, as on border with the Ottoman Empire [15].

This backdrop of war also helped to keep alive the old systems of slavery along the shores of the *Mare Nostrum*. The decline of the ancient world and the emergence of settlements in these open frontiers created diffuse forms of bondage or dependency of the settlers towards their feudal lords [16,17]. Slave labour continued to be an important element in domestic economies of urban regions [18–20]. However, as Marc Bloch pointed out, exceptionally, slavery only remained in rural communities—e.g., in some feudal border regions with the Islamic and Turkish worlds in Italy, southern France and the Iberian Peninsula. *"But even there –unlike in certain Iberian regions, such as the Balearic Islands– servile merchandise was too scarce and too expensive to be used significantly in farm labour"* [1] (p. 245). The exceptional permanence of slave labour until the 16th century also explains why in some Mediterranean regions slavery was a first trial of the sugar plantation system, which was later moved by the European colonizers to Atlantic and African islands [21], and then become established in the American colonies [22,23].

The island of Mallorca constitutes in this regards an original two-way transition process. Firstly, after the Conquest the failure of the Catalan feudal colonisation promoted slavery labour. From 1229–1234, following the invasion, the Andalus population was enslaved in great numbers [24–26]. The new feudal lordship and landownership tried out the combination of two settlement models: autonomous family-run peasant farms and slave gangs who worked on the manorial estates. In the second half of the 13th century slave imports became more regular and intense [27,28]. This was partly because the regular flow of free labour emigration from the mainland was slow and scarce [29]. The reasons were twofold. On the one hand, the consolidation of serfdom in the northeast of Catalonia at the beginning of the 13th century restrained or even prevented peasants from emigrating [30]. On the other hand, the conquest and colonisation of Valencia began in 1237, and it also required peasant migrants for the agrarian colonisation [31]. Thus, after an initial period of hesitation, the resort to slave labour became an increasingly desired option for the local nobility due to the slow growth of the free population in Mallorca. In the second half of the 13th century, Catalan settlers soon realised that isolation was not merely a risk, but it also provided the opportunity to capture and keep slaves [32]. In practice, the island became an authentic field of experimentation in the management of slave labour. This included slave gangs working on the large farms, domestic slavery, and even the hire of slave labour to third parties. In the 14th and 15th centuries slave labour was extensively used in public works and on rural estates.

Sencondly, from 1450 onwards, Mallorca's historical trajectory diverged from other regions with which it shared the same political and legal framework, such as Catalonia and Valencia [33]. The defeat of peasant movements of 1450 and 1520 led the small and medium peasant holdings to become the main actors of the organisation of rural space and agricultural growth in Catalonia and Valencia [34–36]. Conversely, in the Island the peasant dispossession and the enlargement of great estates led to a precocious agrarian capitalism. The two peculiar features of this Mallorcan transition took place simultaneously in the labour market (from agrarian slavery to a low-wage labour market), and in agricultural specialisation (from a former polyculture production ruled by manorial estates and farms belonging to the aristocracy to livestock rearing and, later on, cereals and olive groves). These interlinked changes in land and labour exploitation shaped an agrarian system that lasted until the beginning of the 19th century.

The historical sources used are described in Section 2 of this article. Section 3 explains the decline of slave labour. Section 4 outlines the intense changes that took place in land and livestock property, which stemmed from the socioeconomic crisis and political defeat undergone by the Mallorcan peasantry. Section 5 examines the context in which agricultural specialisation on large estates was developed throughout successive stages. Section 6 analyses the characteristics of the new low-wage labour market. In the concluding remarks we summarise the reason why slave labour, as it had been known until 1500, declined rapidly on the island while it flourished in the Atlantic.

## 2. Sources, Dataset and Methodology Used

Two types of documentation have been used in this study. On the one hand, the quantitative data previously published on the main socioeconomic trends from the 14th to the 16th centuries on Mallorca has been collected and systematized: series of tithe payments and wheat prices from the Royal Archives [37–39]; masons' wages from the Cathedral of Palma [40,41]; and the price and lease of slaves [42–45]. The indicators developed from these series are presented in the appendices. In addition to these series, we have compiled a large sample of probate inventories belonging to the island's aristocracy, composed by estates of the nobility, merchants and the urban patriciate. These sources come from notarial document collections (PN), the court administration (AA) and private archives belonging to the nobility currently deposited in the Archive of the Kingdom of Mallorca (ARM) and the Mallorcan Chapter Archive (ACM).

*Post-mortem* inventories had been extensively used on the island to study private libraries [46,47], material living conditions and consumption patterns [48,49], the evolution and composition of the estates of wealthy groups on the island, and the changes in factors of production on farms: animal power, lands, slave labour, seeds, etc. [45,50,51]. These probate inventories also described the patrimonial assets of the firstborn: real estate (lands, houses, mills, etc.), household goods (furniture, valuable objects, books, etc.), slaves, livestock and stored produce, and rents belonging to the estate (emphyteutic land leases, mortgage loans and public debt). The rights corresponding to the rest of the heirs, and the benefits or legacies bequeathed by the deceased to other individuals or institutions, were calculated on the basis of the monetary value of those assets.

We have used two sources to estimate the impact and geographical distribution of the intense process of land grabbing unleashed in the 15th century. The first is a register of properties purchased from the aristocracy to the indebted peasantry (Table 1). This record (called '*memorial*') was compiled for a judicial process that took place between powerful estates and the peasants of the towns and villages in 1509–1511 [52,53]. The document recorded the number of real estate properties purchased in each rural parish by the aristocracy living in the city of Palma. It contained buyers' and sellers' names, as well as their status (noblemen, merchants, wealthy peasant farmers, etc.) and the type of estate. Unfortunately, it did not include the size of these possessions. The document only distinguished qualitatively between large estates ('*alqueries*'), medium-sized farms ('*rafals*'), smallholdings (vineyards, land plots, gardens or orchards), and houses and mills. In total, over 600 certifications of purchases from rural landholders by wealthy elites from the city of Palma were recorded. Three hundred and ninety of these were large estates (61%), and the rest smallholdings (24%) and houses and mills (15%).

Our study is focused on the large estates acquired by the aristocracy: big estates ('*alqueries*') and medium-sized farms ('*rafals*'). In many cases the purchase and sale statements of large estates did not provide details on whether they had already been the result of previous purchases or mergers of other large and medium farms, so that landownership changes were far greater than what was recorded in this document. This information has been compared with the data provided by the set of probate inventories assembled: 265 large farms ('*possessions*') for the period 1375–1575. Most of these farms had been acquired during the 15th century, which coincided in time with the list of properties recorded in the 1511 '*memorial*'. However, some estates might be recorded more than once in successive inventories of the same family, biasing the dataset. The farms of the inventory sample have been differentiated following the same criteria used in the 1511 source (Table 3). Our sample of 121 inventories of 265 farms

in the period 1375–1575 is quite unusually large for medieval and early modern times (Appendix data). It has been compiled focusing on the inventories of the class of rich landowners, composed of the former lineages of the small feudal nobility, the urban patriciate (*ciutadans*) and wealthy merchants [54]. These groups were the key actors in the appropriation and accumulation of lands, the reorganisation of the agrarian territories that created large consolidated farms known as '*possessions*' in the island, and labour exploitation, which took place between 1400 and 1575. Accordingly, we have not included in the sample the inventories of some remnant wealthy peasants or of merchants and artisans who owned some slaves but not many land.

　　To give a reference about the proportion that our dataset of 120 probate inventories represents out of the total, it is worth mentioning that the Mallorcan feudal nobility and the urban patricians constituted only a small part of the population that lived in the city of Palma. The records of feudal lords and noblemen written at the end of the 14th century listed some 80 heads of family with a noble title. At the beginning of the 16th century only about 60 of these lineages still possessed manorial domains, although the members of the nobility had increased in the meantime. Table 1 compares our sample of probate inventories with the 1511 Register. In the land purchases list of 1511 there are 219 individuals with their names and social status, 87 of whom were noble, and 78 were patricians with another title of distinguished citizens assimilated to nobles. In our sample of inventories we have collected information on 120 estates and 266 farms belonging to 72 families, of which 44 were members of the nobility and 20 of the patrician status (*civis*) and 8 of merchant status. Therefore, the dataset of inventories compiled is a fairly representative sample of the aristocracy of the island. Note that the composition and geographical distribution of the noble estates that make up this sample is very similar to that recorded in the 1511 list of land acquisitions made by the ruling social groups of Mallorca in the 15th century (Table 1).

**Table 1.** Coverage of the Mallorcan nobility and patriciate in our sample of probate inventories (1375–1575) and in the Register of Farms purchased by the aristocracy in 1511.

| | Sample of Probate Inventories | | | Register of 1511 | |
|---|---|---|---|---|---|
| | **Number of Inventories of Estates** | **Number of Family Lineages** | **Number of Farms** | **Number of Family Lineages** | **Number of Farms Purchased** |
| | **1375–1575** | **1375–1575** | **1375–1575** | **1511** | **1511** |
| Noble (*miles*) | 79 | 44 | 173 | 87 | 192 |
| Citizens (*civis*) | 32 | 20 | 76 | 78 | 115 |
| Merchants et al. | 9 | 8 | 17 | 54 | 59 |
| *Total* | 120 | 72 | 266 | 219 | 366 |

Sources: Farms purchased in the period 1509–1511 by the upper social groups throughout the 15th century, reported by the Register of Civil Litigation. [52,53] (p. 152–202). Sample of probate inventories compiled from Notarial Sources. See the Supplementary material.

### 3. Feudal Colonisation and Slave Labour in Mallorca (1229–1500)

After feudal colonization the island was populated with migrants from Catalonia and was settled in King land domain and manorial lands. The peasant's farms were located around villages that evolved towards agro-towns with an increasingly complex social structure. In the top there were the Manorial Lords that stayed in the island after the 1229 conquest, and bellow the settlement process enlarged a group of wealthy peasants that owned large farms, followed by another group of medium and small peasants who owned medium-size tenancies or small plots. At the lowest ranges there were the journeymen and slaves. Prior of 1400 the bulk of the rural families (more 70%) were medium peasant owners or small holders with petty tenancies or plots. However, the tax records dating back to 1328 recorded 2800 slaves owned by wealthy peasants, and the ones of 1428 still recorded more than 1100 slaves living in intermediate farmsteads. Our calculations suggest that this slave population owned by peasant families could have represented about 15% of the rural population [19,27], without counting the large contingents of slaves owned by the big urban and rural property owners. This means that the slave population represented a fairly large percentage of the agricultural labour force until the middle of the 15th century [45,50,55].

According to the number of households registered in the capitation tax ('*morabatí*') records, the effects of the Black Death and the successive waves of epidemics and famine between 1329 and 1421 led to a 32% decrease in the island population. The crisis was more intense in urban districts, which suffered the loss of half of their households. In villages, around 22% of their inhabitants were lost. In 1329 the population of the island was 56,000 inhabitants, of whom 56% lived outside the city of Palma; in 1364 it was of 48,800 inhabitants according to the records of the *morabatí* tax, and the subsequent plagues, epidemics, famines and the civil war made decrease the population to 31,000 in the middle of the 15th century. The population only recovered in the second half of this century reaching 52,000 inhabitants [56–59].

Although the effects of the epidemic cycle were not as devastating as those documented in other European regions [60], it would have been enough to disrupt, albeit temporarily, the agrarian class structure and, most especially, the labour market. During the population slump in the second half of the 14th century the coexistence of the two labour markets increased tension and conflicts. And they, in turn, led to contradictory policies on the control of free and slave labour. The decline in labour supply resulted in a sharp rise in urban and rural wages [55]. The authorities' attempts to tackle these problems were as contradictory as those described for other European regions [61]. First they tried to attract labour, then they wanted to control its mobility, and finally they sought to avoid an increase in nominal wages. Several regulations were established after 1350 labour by-laws that forbade piece-rate contracts and established that salaried workers had to be hired on a daily basis. However, simultaneously, attempts were being made to control the rise in urban and rural wages [59,62,63]. However, the market requiring the greatest regulatory effort was that of slave labour. Studies on the second half of the 14th century have shown that slaves played a fundamental role in the reorganisation of work on farms and manorial estates, as shown by inventories and tax records dating from the second half of the 14th century and the first half of the 15th century. Slave gangs were a fundamental element in the land cultivation of the island.

How worked slavery in the island from 1350–1500? The most relevant features of Mallorcan slavery during de 15th and 16th centuries were:

(1)　First, obtaining slave labour did not depend on an internal reproduction process. This is shown by the high rate of male slaves in the slave gangs recorded in the landowners' inventories and purchase-and-sale contracts. After the Black Death, slave imports represented a serious competition for free wage labour. Imports from the Levantine slave markets became the flow that supplied the internal demand. Until the end of the 15th century the slaves imported were in 80% young men coming from the Black Sea and recorded as Russians, Tartars or Muslims. Only from then on there appeared black men coming from Africa, but they represented only about 18% of the total slaves sold [27,44,63,64].

(2)　In addition, the local authorities provided maritime and military facilities to this slave trade and surveillance and the market requiring the greatest regulatory effort was that of slave labour. The royal authorities also facilitated the drawing up of commercial self-rescue contracts (known as '*talla*' and '*alforria*' in Catalan), through which slaves could buy their own liberation when they got older, their work performance was expected to start to decline and their purchase price had already been amortised by their owners [64–67].

(3)　The administrative and judicial system of the Kingdom of Mallorca provided legal cover for the system of slave control, which was in the hands of officials from the royalty and nobility. The first legislation was introduced in 1328 to control slave mobility, and to compensate the owners of fugitive slaves. Different regulations regarding the safe-keeping and custody of slave labour were drawn up by the authorities. The numerous laws and regulations passed by the Kingdom Assembly, or by the Crown of Aragon town councils [44,64], can only be compared to those passed in Barcelona [68]. The oldest one dates from 1354, and was followed by a long list whose scope could range from the city of Palma to the entire Kingdom until 1500. In addition, several local laws were passed at parish level, especially in coastal regions. At times, provisions regarding slaves were included in by-laws of a more general nature [63] (pp. 65–70). These rules established harsh punishments for fugitives, and for free men who facilitated their escape, which ranged from flogging –up to 100 lashes— to the amputation of limbs, and even the death penalty. Slave mobility and any kind of relationships with free citizens were also restricted. The monitoring of slaves control led to the restoration in 1380 of the watchman ('*mestre de guaita*'), whose job thereafter consisted of slave surveillance [69].

(4)　The Kingdom of Mallorca authorities also ensured the right of owners to mistreat their slaves, with hardly any limitation. Although some specific requests were not explicitly accepted by the kings –considering they could encourage crime— they were implicitly tolerated. The owners' right to 'correct' their slaves' behaviour by mistreatment ('*tam corrigendo vel castigando*' [either amending or correcting]), especially by flogging, was confirmed although restricted *de facto* to slaves [70] (pp. 32,163,184). The development of this penal legislation was accompanied by procedures to compensate the owners of fugitive slaves, particularly those whose slaves had died during the persecution, or were later executed [63] (pp. 65–70). In short, the authorities definitively helped to ensure slave owners their livelihoods by providing them with all possible means to secure the control of the big slave prison in which the island of Mallorca had become. The consolidation of different forms of slavery, agrarian and urban, would not have been possible without the active policy of the Mallorcan authorities. The Kingdom's judiciary power and the parliament were dominated by social groups made up of slave owners—i.e., gentlemen, the urban patriciate and also wealthy peasants. For them, ensuring the correct functioning of slavery was a fundamental issue. Yet, despite the major development in laws and regulations, non-compliance was also very common. In fact, the 1374 by-laws were either not complied with or repealed at the request of the island's representative assemblies. The 1390 laws were repealed by the king himself, and the 1393 laws were repealed in 1401 in exchange for new taxes imposed by the king. Minor provisions relating to agrarian practices, such as the imposition of night imprisonment, were subject to transgressions with the owners' tacit consent. Indeed, the night imprisonment order hampered the duties of shepherding, one of the slave occupations, and was not adhered to at the request of the owners themselves [50,63,64].

Table 2 shows some indicators that reveal the economic conditions in which the modes of slave labour –direct, slave gang hire, and under self-liberation systems— were made possible during the first half of the 15th century. It displays the average price of slaves sold in Palma market in ten-year averages, and the price (on a daily wage basis) of hiring individual slaves per day, both performing the same work in the Cathedral of Palma, compared to the nominal daily wage of a free day labourer. The evolution in the number of households paying the capitation tax ('*morabatí*') is also given in total numbers and only for the rural regions, which can be taken as a proxy for demographic trends.

**Table 2.** Slave prices and wages of free daily labourers working as builders (100 = 1391–1400).

| | I Price of Slaves | II Rent of Slaves Hired on a Daily Basis in Building | III Wages of Free Day Labourers in Building | IV Slave Hiring/Wages of Free Day Labourers in Building | V All the Households Paying the Capitation Tax ('*morabatí*') | VI Rural Households Paying the Capitation Tax ('*morabatí*') | YEAR of the Tax Registers used to Account Households |
|---|---|---|---|---|---|---|---|
| 1390–1400 | 100.0 | 100.0 | 100.0 | 100.0 | 100.0 | 100.0 | 1364 |
| 1401–1410 | 108.5 | 98.3 | 127.4 | 77.2 | n.a. | n.a. | n.a. |
| 1411–1420 | 126.8 | n.a. | 127.4 | n.a. | 78.7 | 96.6 | 1421 |
| 1421–1430 | 141.8 | 122.8 | 124.2 | 98.9 | 74.7 | 93.9 | 1427 |
| 1431–1440 | 149.1 | n.a. | n.a. | n.a. | 63.4 | 81.2 | 1444 |
| 1441–1450 | 145.5 | 102.3 | 112.1 | 91.3 | 63.8 | 79.1 | 1451 |

Sources: Our own, data wages was taken from 1390 to 1430 from [40] (pp. 75–100) [41] (p. 43); and for 1441–1450 from ACM 1727–1733 (labourers' and masters' wages on a daily basis). Data on slave prices was taken from [43–45].

Between 1364 and 1451 there was a steady population decline in Mallorca (columns V & VI in Table 2). The scourge of epidemics and famines continued to hamper demographic recovery in rural areas (where the population diminished 21%) and, even more, in the city of Palma (where the population decreased 36%). At the same time, the price of slaves continued to rise throughout the first half of the 15th century (column I in Table 2) reflecting the high demand for slave labour in the island despite the increasing difficulties in the Levantine slave markets in the Black Sea and the East Mediterranean region where Ottoman pressure was disrupting the capture and trade. Against this backdrop of labour shortage, "*slavery was the most efficient means by which the ambitious and powerful could become richer and more powerful. It was the answer to energy shortage. Slavery was widespread within the somatic energy regime, notably in those societies short on draught animals. They had no practical option for concentrating energy other than amassing human bodies*" [71] (p. 12).

The relative abundance of slave labour competed fiercely with free wage labour during this period of demographic decline in Mallorca. In an ordinance dating from 1393, the Governor complained about the population decrease that had taken place on the island and attributed the lack of immigrants to the fact that they could not find work due to the excess ('*gran multitud*') of slaves that made up the workforce [63] (p. 62). Our data endorse the concerns of the authorities. Throughout the first three decades of the 15th century the daily wages of building workers rose, and the scarce data available on agricultural labourers' wages point in the same direction. However the price of free day labour increased up to 27%, while the price of slaves hired per day only grew by 22%. Then, during the two decades before the Mallorcan Civil War of 1450–1454 the price of both free and slave labour decreased, but the former more significantly than the latter (columns II & III in Table 2).

Thus, from 1375 to 1450 the relative price of hiring slave labour became more profitable than free wage labour in Mallorca (column IV in Table 1). Moreover, the slaves themselves had an incentive to participate in this labour market, since it was their only way to obtain cash to buy their emancipation in a way that also enabled the owners to maximise their investment. The increase in slave prices and the simultaneous decrease in slave hire during this period (columns I & II in Table 2) can be explained by the combination of two phenomena. On the one hand, speculation with the purchase and sale of slaves in a labour-scarce market; and, on the other hand, the proliferation of self-liberation contracts indicated that the owners were willing to sell older slaves once their investment had been amortised. The combination of both activities would explain the relative convergence of slave hire with the price of free labour. If trends on the island of Mallorca were already moving towards the new patterns detected in Iberian slave markets is something that deserves further research [72].

The *post-mortem* inventories provide reliable information about the changes that took placed in the 15th century. In the last quarter of the 14th century the exploitation of slaves had been a way for the large and medium landowners to avoid depending on a wage workforce. Around the year 1400, slavery was at its peak on the island, and slave labour was used across all economic sectors from agriculture and manufacturing to construction and services [40,41,63].

The sample of probate inventories studied for the period 1375–1400 gives similar results to other sources studied for the second half of the 14th century (Table 3). The rich urban landowners and aristocrats had an average of 7.4 male slaves and 1.6 female slaves on their estates while the wealthy peasants possessed an average of 1.5 slaves per owner [50]. The data from the first half of the 15th century (1401–1450) indicates an increase in the number of both male and female slaves with respect to 1375–1400. The average of the total number of slaves per estate, and per farm, increased from an average of 9 to 11.5. The difference between the average per estate and per farm responds to how the amount of work on those farms was distributed. If the main purpose was agricultural, the number of slaves was highest. Contrarily, if the farm units were devoted to livestock rearing there were few slaves, if any. Moreover, the proportion between male and female slaves shows that male slaves were set to work on agricultural tasks where physical strength was required, while female slaves were destined for domestic service in the farmsteads. This allocation of slave labour was consistent throughout the period studied. From the second half of the 15th century onwards, as the agrarian estates grew in

size and their specialisation in livestock intensified, the number of slaves per property and per farm went down to a total of 8 slaves in the period 1451–1500, and 6 during the first half of the 15th century. By the middle of the 16th century rural slavery had practically disappeared in Mallorca.

**Table 3.** Average number of slaves owned by aristocratic estates and farms.

| Slaves/Estate | Number of Estates | Male Slaves | Female Slaves | Total Slaves |
|---|---|---|---|---|
| 1375–1400 | 11 | 7.4 | 1.6 | 9.0 |
| 1401–1450 | 12 | 9.1 | 2.4 | 11.5 |
| 1451–1500 | 24 | 6.4 | 1.5 | 7.9 |
| 1501–1550 | 55 | 4.1 | 1.1 | 5.3 |
| 1551–1575 | 18 | 0.9 | 0.4 | 1.3 |
| **Slaves/Farm** | **Number of Farms** | **Male Slaves** | **Female Slaves** | **Total Slaves** |
| 1375–1400 | 17 | 4.8 | 1.1 | 5.8 |
| 1401–1450 | 23 | 4.7 | 1.3 | 6.0 |
| 1451–1500 | 47 | 3.3 | 0.8 | 4.0 |
| 1501–1550 | 109 | 2.1 | 0.6 | 2.7 |
| 1551–1575 | 70 | 0.2 | 0.1 | 0.3 |

Source: Our sample of inventories, 1375–1575. Sample of probate inventories compiled from Notarial Sources (see the Supplementary material).

After the year 1500 the new farms acquired or created by subdividing larger estates were rented together with part of the working animals, and the seeds for cultivating cereals, but without mentioning slaves. In the middle of the 16th century, the estates of the aristocracy were comprised of a series of farms exploited by lease-holders contracted from the agro-towns, who in turn hired a wage workforce to cultivate the fields and pasture the livestock. Slavery was circumscribed to the sphere of domestic service. At the end of the century very few farms still had slaves as part of their agricultural workforce [73].

Paradoxically, in the second half of the 15th century new regions in West Africa and the Gulf of Guinea opened up to the capture and sale of slaves. According to notarial registers, the price of slaves went down in the Palma market (Figure 1). These new slave trading emporiums broke the stranglehold held by the eastern slave market [72,74–76]. This new flow of slaves to the Atlantic and Mediterranean cities of the Iberian Peninsula, such as Seville and Barcelona, was continuous and abundant [77–80]. Up to 1440 there were still important opportunities for the use of slave labour, either in agricultural or urban sectors. And, in addition, new business opportunities were opening up with the growing slave emporium in the Gulf of Guinea [72,81]. The hiring of slave labour was still a more attractive alternative than free labour before 1450, but something started to change from then on. Although the island's purchase-and-sale registers for slaves recorded the arrival of the new sources of forced labour, the number of purchases registered by notaries started to decrease, as did the average price of slaves (Figure 1). Therefore, despite the growing supply of slaves there are clear indications that demand for them was diminishing in Mallorca.

This decline came about at the same time as novel modes of slavery were being developed, linked with new demands for tropical goods and the fresh flows of slave labour from West Africa. The new slave plantations that were being experimented with in the Atlantic had some distinguishing features from the slavery of classic and mediaeval antiquity. The new plantation model based on slave gangs was linked with the exportation of the new colonial products—sugar, coffee and tobacco—demanded by ascending intermediate social groups in Atlantic Europe, the heart of a new colonial capitalism in expansion [82,83]. However, after the great peasant revolt of 1450–1454, social and agrarian transformations were taking a different direction in Mallorca. In economic and social terms, slave labour ceased to be a relevant alternative for the island's new landowning class.

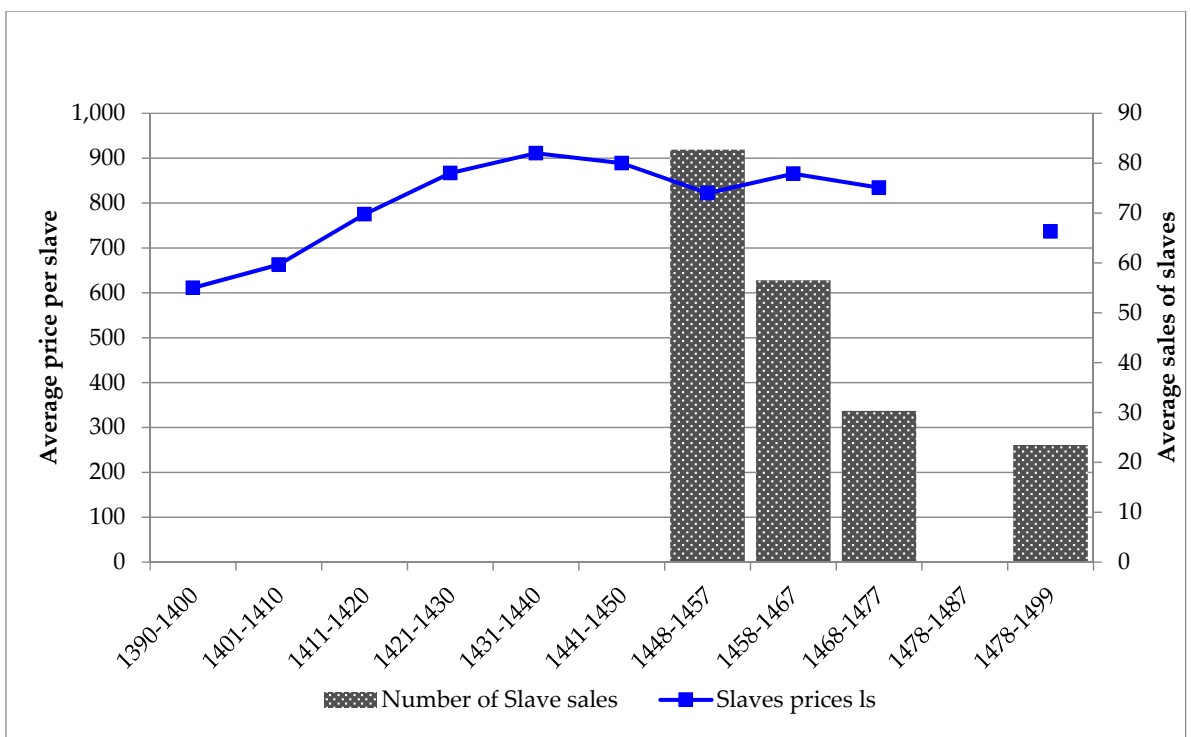

**Figure 1.** Prices (1390–1477) and sales (1448–1499) of slaves in Mallorca. Source: Authors' own elaboration from [43,44] and [45] (pp. 19–22). The number of slave purchases and sales, and the average prices of slaves have been calculated from notarial sources.

## 4. Enclosures, Land Appropriation and Estate Enlargement (1375–1575)

After the Black Death the small plots of land that had become vacant due to the death or abandonment of their owners were resettled by peasant families, whenever possible, under the direct rule of lords or landowners who had to resign themselves to lower rents. Even so, approximately two decades after the 1348 plague there were still plots that had not been re-cultivated [84]. Other signs that the pressure on lands had decreased was the fall in the property tax rate on inheritances, purchases and sales ('*lluïsme*'), and the decrease of land rents (emphyteutic censuses) paid by settlers on vacant lands [37] (pp. 64,65). The evidence also points to the concentration of lands. Many of the lands that had been broken up in settlement processes prior to 1340 were regrouped in larger farms.

Until 1400 the land was mainly in hands of peasant families, and only the manorial demesnes and some rural properties around villages and the city of Palma were in the hands of the landlords, merchants and patricians [51,84,85]. This wealthy peasant society was disrupted after the military and political defeat of the plebeian uprising and civil war (1450–1454), and an intense and wide process of land appropriation took place against a backdrop of different economic and social conflicts. On the one hand, the increase in tax burden since the late 14th century led to a growing indebtedness of rural townships, villages and peasant holdings. People were forced to sell their slaves, livestock and lands to pay off their debts [86–89]. In 1450 the peasants rose up against this process of impoverishment. After a civil war (1450–1454), the Mallorcan peasantry was defeated and subjected to extremely harsh political, military and economic repression. Furthermore, the nobility failed in their endeavour to restore their incomes by increasing feudal rents, and the urban patriciate met strong opposition to their attempts to increase their revenue by collecting the king's rents and public debt [90–92].

The peasant property crisis provided the emerging ruling groups with new investment opportunities. The nobility made use of their jurisdictional powers, and patricians of their capital, to appropriate peasant holdings. This land grabbing process took place in successive waves of dispossession and accumulation. The first wave was led by the powerful rural elites, who acquired their neighbours' lands. A second wave of land purchases through debt execution was carried out

when patricians and merchants of the city of Palma acquired many fiefdoms that had once belonged to former noble lineages then bankrupt, together with many farmhouses that belonged to farmers in financial straits. Lastly, the aristocrats who had been able to overcome indebtedness (feudal lords and entitled nobility) joined this process by expanding their domains through the acquisition of adjacent lands. These appropriation processes were also extended to common land, and to the rights of common use over woods and pasture lands belonging to villages. The peasants' social and political protests failed to reverse this dramatic change in land property rights, nor did the subsequent profound reorganisation of land uses, agricultural landscapes, and labour exploitation [90–95].

Table 4 shows that the two sources provide similar results in terms of which social groups were the key actors in the land grabbing, although the 1511 register shows a wider social range. According to this data, the landed aristocracy and the urban patricians were the two most relevant social groups acquiring large farms to add to their estates throughout the 15th century. The two together acquired more than 80% of the total number of large farms in the 1511 report, and also more than 80% of the rest of the real estate sold by the peasants (plots of land, houses and mills). The large estates studied in the set of inventories provide similar figures: more than 80% of the farms listed in the sample were in the hands of aristocrats and patricians, while merchants owned a smaller proportion. It must be pointed out that many of the families that were members of the merchant class in 1400 had risen socially by through having acquired fiefdoms and lands throughout the second half of the 15th century, first becoming members of the urban patricians and joining the ranks of the aristocracy [52,53]. Last, Table 4 suggests that these social classes preferred buying large compact farms. This was a distinctive feature of the process: the aristocracy and the patricians were not interested in acquiring land as a renting strategy but to control large, compact extensions of land [91].

**Table 4.** Sample of probate inventories of large estates (I 1375–1575) compared with the register of farms purchased in 1509–1511 (R 1511) by wealthy social groups living in the City of Palma.

| Social Groups | 1511 Register of Farms Bought % | 151 Register of Vineyard Plots Bought % | 1511 Register of Houses & Mills % | 1511 Register of Total Purchases % | 1375–1575 Sample of Inventories of Large Estates Inventories % |
|---|---|---|---|---|---|
| Aristocracy | 51.4 | 45.2 | 47.2 | 49.4 | 64.8 |
| Patricians | 32.4 | 34.9 | 28.1 | 32.4 | 28.7 |
| Merchants | 3.9 | 11.0 | 4.5 | 5.6 | 6.5 |
| Professionals | 7.2 | 5.5 | 9.0 | 7.1 | - |
| Craftsmen, lawyers & notaries | 5.1 | 3.4 | 11.2 | 5.6 | - |
| TOTAL | 100.0 | 100.0 | 100.0 | 100.0 | 100.0 |

Sources: Register 1511: Farms purchased by social groups throughout the fifteenth century, reported by the Register of Civil Litigation [52,53]. Inv. 1375–1575: Sample of inventories compiled from Notarial Sources data and sources of probate inventories (see the Supplementary material).

Table 5 and Figure 2 contain information about the location of the large farms acquired by the aristocracy in the different agro-ecological districts of the island. The first two columns show the land extension of the district with respect to the total rural land (without including the City district) and the population densities of each district. The following two columns give the percentage of farms acquired by the aristocracy in each agro-ecological area, according to the 1511 register and the sample of inventories. These districts had been formed during the mediaeval agrarian colonisation of the island [24,96]. Pla region in the centre of the island was the second largest district in terms of land extension with a population density above the average for rural Mallorca. Until 1500 it was mainly comprised of large and medium farms devoted to cereal crops and extensive livestock rearing, with settlements forming large agro-towns surrounded by a crop-intensive belt of vineyards and horticulture small plots. This thick mosaic of vegetables gardens, irrigated orchards and vineyards was especially evident on the Raiguer land that stretched down to the Albufera wetlands of Alcúdia bay [84,94].

**Table 5.** Aristocracy, patricians & merchant owners of large estates.

| Regions of the Island | % of the Mallorcan territory | Inhabitants/Km$^2$ | % Distribution of Large Land Ownership in the 1511 Register | % Distribution of Large Land Ownership in the Inventories 1375–1575 |
|---|---|---|---|---|
| Llevant | 16.1 | 7.4 | 14.7 | 3.1 |
| Migjorn | 23.9 | 8.7 | 18.2 | 32.6 |
| Pla | 21.9 | 14.3 | 45.8 | 33.0 |
| Raiguer | 9.9 | 24.1 | 5.4 | 5.7 |
| Tramuntana | 28.2 | 11.7 | 15.8 | 25.7 |
| Total | 100.0 | 13.2 | 100.0 | 100.0 |

Sources: Population from *Morabatí* Tax [58]. Register 1511 and Probate Inventories 1375–1575 as in Table 3.

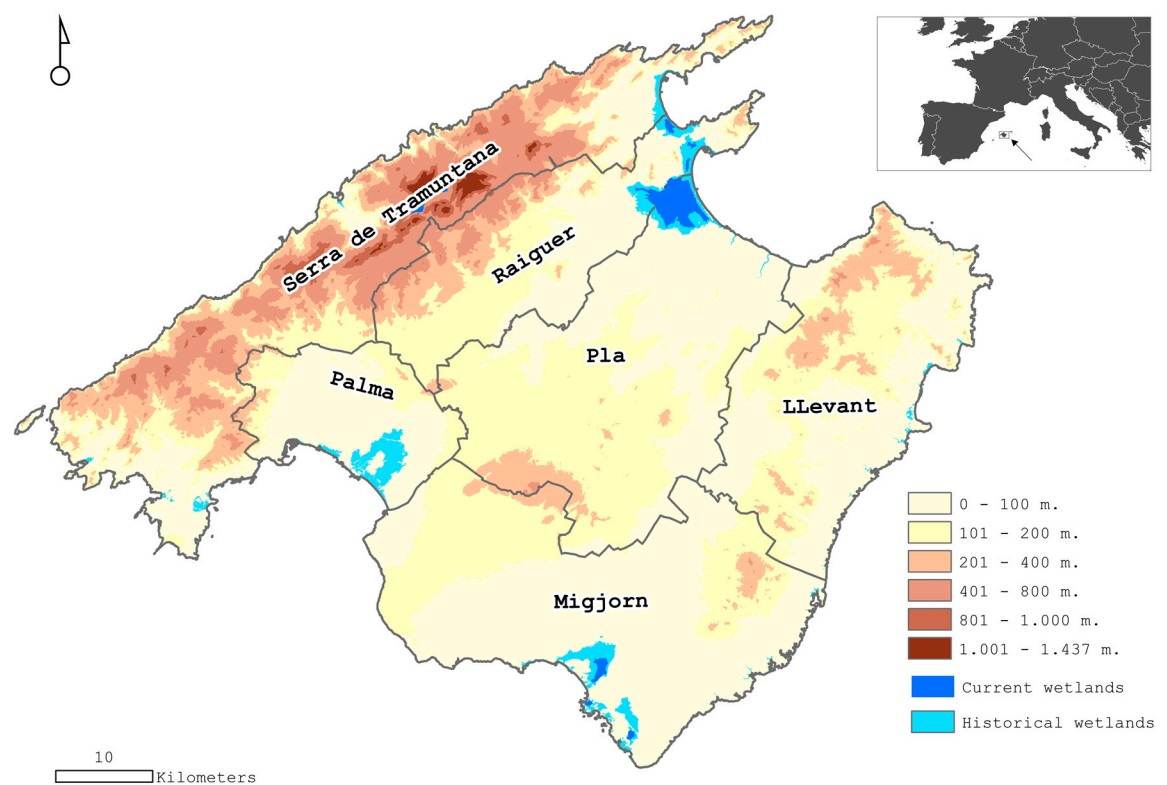

**Figure 2.** Map of the Mallorcan regions. Source: Our own.

In the arid southeast and northeast districts of the island, Migjorn and Llevant respectively, the land was poorer and population densities were the lowest on the island. This was where extensive livestock farming was developed. In the Tramuntana Mountains the population density were somewhat lower than the island average, and was organised into small hamlets located in the valleys or at the foot of the mountain chain. There the few flat lands were used for polyculture production and vineyards, while the mountainsides and peaks were extensively occupied by medium and large farms. The peaks were used as summer pastures for the herds that spent the autumn and winter on the estates in the centre and east of the island. Steep lands were terraced to grow olive trees and the vines associated with cereals. Last, there was Raiguer region located between the mountain chain and Pla, with fertile lands and higher population densities, dispersed settlements and intensive peasant agriculture based on vineyards [96–99].

According to the 1511 register, 45% of the purchases made by the new aristocracy, and 33% according to the sample of inventories, were located in Pla. The Migjorn area (18%), Llevant (15%) and the Tramuntana mountains (16%) were also subject to the land grabbing at the hands of the new merchant aristocracy. Raiguer region, however, was relatively unaffected (5%). The sample of inventories confirms that the properties acquired were mostly located in Pla (33%), Migjorn (33%) and Tramuntana (26%) where population densities were lower. This accumulation of formerly

medium-sized peasant farms ('*rafals*') created much bigger, compact units of between 500 and 4,000 hectares, which included agricultural farms in the inner and eastern districts, and extensive properties on the mountainsides and peaks of the Tramuntana mountain chain [97,100].

We can infer from the available sources that these land purchases were also forced sales due to their previous owners' indebtedness. Credit troubles and insolvency not only affected a large number of smallholders and formerly wealthy peasants, but they also comprised vast stretches of land belonging to the town councils. Most of the new estates were a combination of purchases of large farms in flat regions, both in the centre and to the east of the island, and land on the mountain chains of Tramuntana and Llevant. There was certain logic behind this purchasing strategy; it arose from a very specific economic trend that would mould the new agrarian landscapes of the island throughout the 15th century.

## 5. Livestock and Agricultural Specialisation after the Failure of Sugarcane Slave Plantation in Mallorca (1375–1575)

What were the agro-economic objectives behind these extensive estates created throughout the 15th and 16th centuries? Slavery had found a new purpose with the development of sugar cane production in the 15th century, and the first enclaves were developed in the Mediterranean [101]. In the southeast of the Iberian Peninsula, sugar exploitation models based on peasant farms under manorial coercion were also being experimented with in Granada and Valencia [102,103]. However, the unsuitable agro-ecological conditions –as sugar required a lot of fuel and soil nutrients— and the costs of labour supply limited their expansion [104] (pp. 19–61). The sugar frontier moved over to the Atlantic (Madeira, the Canary Islands, the Gulf of Guinea) where both the soil and climate were more suitable, the natural resources more abundant, and new slaves were captured nearer by. The first sugar plantations with a slave workforce were tried out on these islands [105].

The attempt to introduce this sugar plantation model on the island of Mallorca also failed. In 1464, a company made up of merchants and landowners bought a large area, amalgamated all the farms to cultivate sugar cane ('*canyesmel*') and built a sugarcane mill ('*trapitx de sucre*') [106] (pp. 177,178). The company asked the Governor for the privileges of disposing of all the available water to work the mill, and to be exempt from paying taxes on their activity. Between 1466 and 1470 capital increases were made to improve the production capacity of the company. Two slaves were included in the inventory of this estate, in addition to others who were privately owned by the partners of the company, together with the sugar mill, the boilers to obtain the molasses from the cane, the draught animals, and 500 sheep to provide enough manure. The clauses of the company contracts stated that if the economic activity ceased due to a lack of water, or an infestation of worms ('*multitud de cuc*'), the lease-holders would not have to pay the rent for the year prior to the cessation of cultivation. Indeed, sugar production stopped completely in 1478. Plagues and water scarcity were decisive factors in the failure. A new exploitation model better suited agro-ecologically and providing better socioeconomic security was introduced: rearing ovine livestock and commercializing the wool [106] (pp. 179–181).

This example shows, first of all, that the Mallorcan merchants were well aware of the changes taking place in the new Atlantic markets. And second, that this type of plantation could be a profitable alternative to make use of the elements previously accumulated in Mallorca by the new merchant aristocracy: slave labour and land. However, the agro-ecological conditions in this part of the Mediterranean were not suitable enough for this type of production [104] (pp. 19–30). Even in the islands of Madeira, Santo Tomé and Principe, where environmental conditions were apparently more favourable in terms of fuel (wood), water (for irrigating and boiling the cane), work (slave labour) and 'virgin' land rich with soil nutrients, the pressure exerted by slave plantations on the land and the workforce led to the first sociological crises in the new sugar frontiers in the 15th and 16th centuries [107–109].

The agro-ecological unsuitability of this plantation model on the island of Mallorca, the opportunities provided first by extensive ovine livestock rearing and, later on, by exporting olive

oil, steered the development of the large estates newly grabbed from the defeated peasantry [33,89,90]. Analysis of the inventories of our sample (Table 6) together with the evolution in the composition of the king's tithe rents during the period 1375–1550 (Figures 3 and 4), allows us to relate the formation of large estates owned by the new merchant aristocracy with the land-use changes in crop distribution.

**Table 6.** Number of farms and livestock heads per estate in Mallorcan probate inventories of large estates.

| Farms & Livestock Heads per Estate | Farms/Estate | Oxen Draught Heads/Estate | Equines Draught Heads/Estate | Donkeys Draught Heads/Estate | Cows & Calves Heads/Estate | Sheep Heads/Estate |
|---|---|---|---|---|---|---|
| 1375–1400 | 1.55 | 10.0 | 4.0 | 2.9 | 5.1 | 163.5 |
| 1401–1450 | 1.92 | 6.3 | 5.1 | 6.5 | 9.9 | 293.6 |
| 1451–1500 | 1.96 | 6.8 | 3.8 | 3.7 | 8.8 | 303.6 |
| 1501–1550 | 1.98 | 6.8 | 3.4 | 2.4 | 12.6 | 745.3 |
| 1551–1575 | 3.89 | 13.1 | 10.3 | 3.7 | 8.6 | 1213.6 |
| **Farms & Livestock Heads per FARM** | **Farms** | **Oxen Draught Heads /Farm** | **Equines Draught Heads /Farm** | **Donkeys Draught Heads/Farm** | **Cows & Calves Heads/Farm** | **Sheep Heads/Farm** |
| 1375–1400 | 17 | 6.5 | 2.6 | 1.9 | 3.3 | 105.8 |
| 1401–1450 | 23 | 3.3 | 2.7 | 3.4 | 5.2 | 153.2 |
| 1451–1500 | 47 | 3.5 | 1.9 | 1.9 | 4.5 | 155.0 |
| 1501–1550 | 109 | 3.4 | 1.7 | 1.2 | 6.4 | 376.1 |
| 1551–1575 | 70 | 3.4 | 2.6 | 1.0 | 2.2 | 312.1 |

Sources: the same sample of inventories reported in Table 2. Sample of probate inventories compiled from Notarial Sources data (see the Supplementary material).

Table 6 uses as indicators the number of farms per estate and the number of livestock heads per farm and per estate. The data comes from the sample of *post-mortem* inventories and differentiates between the number of draught animals (bovines and equines) and sheep possessed by each type of farm ('*alqueries*', '*rafals*' and '*possessions*'). The number of different livestock heads per estate and per farm provides a good proxy for the ongoing changes in the allocation of productive factors of the large estates. The first column shows the evolution of the number of farms per estate from 1375 to 1600. The ratio indicates that, until the end of the 15th century, the incorporation of new farms into former estates had not modified the management of the previously existing agrarian spaces. Nonetheless, the stability of the ratio responds to two different processes. The first has to do with the structure of the estates at the end of the 14th century and the beginning of the 15th century. At that time, the manorial and urban patrimonies did not generally include more than one large farm unit, be it a manorial estate or a farm on the outskirts of the city of Palma. From the second half of the 15th century onwards, the incorporation of new farming units increased the land possessed in terms of size, while the management remained integrated. The aim of this process of land accumulation and concentration was to create extensive estates managed as large agro-pastoral complexes for animal husbandry, mainly devoted to itinerant sheep grazing. This process took an about-turn in the second half of the 15th century, with landowners fragmenting their estates into new individual farms destined for cultivating cereals and olive tree plantations [51,87–89].

The changes in the composition of herds are explained by these processes related to how the agrarian space was organised. Prior to 1400, the average numbers of sheep per estate and per farm were 164 and 105, respectively (Table 6). In the first half of the 15th century these numbers increased to 294 and 160 respectively, incrementing more slowly in the second half of the 15th century to 304 and 155. The most notable increase came about in the first half of the 15th century when the average number of sheep reached 745 per estate and 387 per farm. The number of sheep per estate continued to rise to 1213 heads in the last period, but the average per farm decreased slightly to 312 heads. These changes can also be appreciated, albeit less intensely, on the bovine rearing farms. In the second half of the 15th century, and the first half of the 16th, the rearing of cows and calves increased (destined for meat production and leather, as well as oxen for tilling). During the same period, however, the ratios for both bovine and equine draught animals remained relatively stable with respect to both estates and

farms. Only in the second half of the 15th century was there a significant increase in the draught power of equines (mules) on the large estates.

The expansion in sheep rearing to produce wool, and in cow and calf rearing for meat and leather, was related to a growing demand from the urban markets and manufactures [110,111]. The stability of the averages for draught animals, on the other hand, reflects the stagnation of cultivated spaces in the crop patterns of the aristocracy's farms. From the middle of the 16th century onwards, the expansion of the cultivated space and the intensification of rotations led to a greater demand for draught animals. The estates were divided into smaller sized farming units and livestock pastures, and the ones with the most draught power and profitable livestock (mainly sheep that provided manure besides wool and meat) were also the ones that specialized in the production of cereals [51].

The changing composition of the tithe rents collected by the monarch across the whole island is illustrative the land use changes made by the large estates of the Mallorcan aristocrats (Figures 3 and 4). The tithe was leased each year for a certain price in cash for each parish. Hence, its internal proportions reflect the changes in the evolution of the nominal income of every agricultural product. The tithe was a levy of 10% of the cereals (wheat, oats, barley and other less cultivated grains) and legumes, and 6.7% of the livestock heads born annually in the different herds (lambs, calves, equines, etc.), with no levies either on wool or other animal products. The tithe on olive trees was a levy of 8% of the olives picked. On the grapes from the vineyards and garden vegetables, the proportion of the tithe was 9.1%, and on flax and saffron it was 7.6% [39,99]. With no agricultural prices available for this period, we cannot deflate the value of the tithe rents collected by the king to obtain a precise quantitative figure for these products in fresh weight. However, the composition of the tithes at current prices is a useful indicator of the ongoing changes in agricultural production, as well as of the ensuing distribution of the nominal agrarian incomes throughout this long period. We must bear in mind that the following figures summarise two vectors in one historical series: the evolution of production and the simultaneous changes in the relative prices of these products.

Figure 3 shows the composition of Mallorcan agricultural production based on the tithe data. It shows that the proportion of cereals in the tithe income was relatively stable until 1550. Other available data on the physical production of cereals for this period show a similar evolution [112]. Therefore, we can safely say that the expansion of cereals, both in physical and in monetary terms, came about in the 15th century and most notably in the second half. The surge in prices and the production of cereals increased the income from the tithes [39,99].

In Figure 4, the grain tithes have been removed to better capture the trend of the new crop specialisations It illustrates how, beginning in the second half of the 15th century, the crops associated with small and medium peasant holdings were being side-lined in both relative and absolute terms. Vineyards were spread across the Raiguer (Figure 2) at the foot of the mountain chain on the axis that crossed the island from the city of Palma to Alcúdia, and around some villages in the centre of the island. Part of the consumption was destined for the local markets, but a large part of the surplus was exported to other regions from the port of Alcúdia. From the middle of the 15th century to the 16th, several factors combined to choke the income from the vineyards. First, the increase in public taxes which, in addition to the tithe, charged the distribution and sale of wine [113–115]. Second, the growing pressure exerted by sheep farming and the privileges enjoyed by the herds in terms of the routes they used on their annual transhumance through the vine-growing regions at the foot of the mountain chain, which damaged the vines. Last, there was the expansion of olive groves on the new estates established on the sides of the mountain chain [96,98].

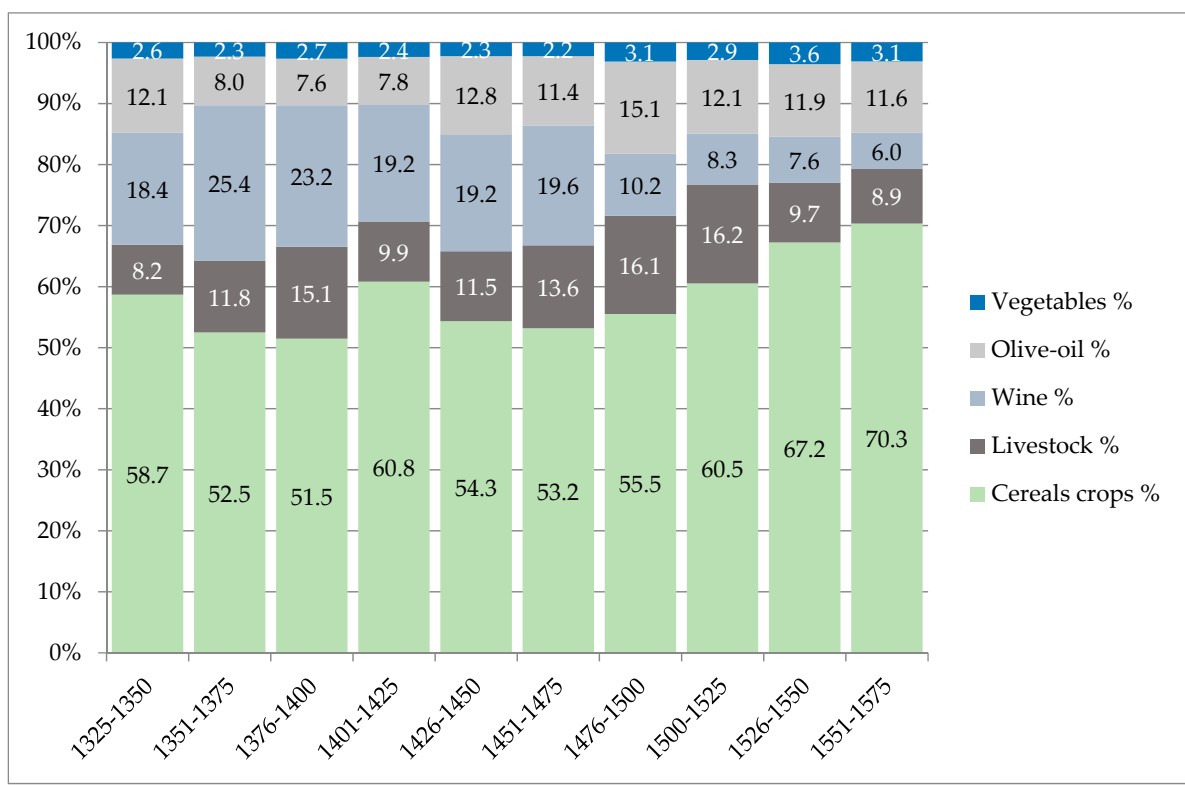

**Figure 3.** Composition of the monetary value of the tithes collected in Mallorca from 1325 to 1575. Source: Authors' own elaboration from data takem from [37–39].

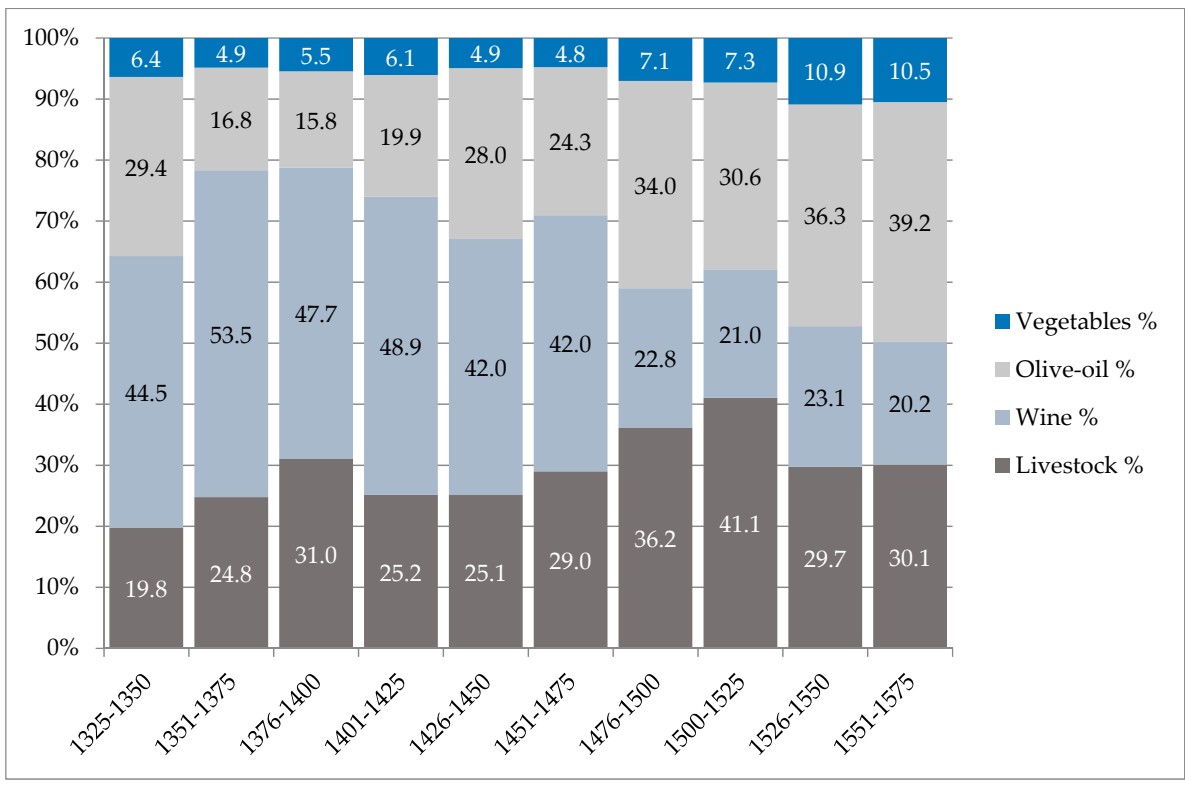

**Figure 4.** Composition of the monetary value of the tithes collected in Mallorca from 1325 to 1575 excluding grains (*ad valorem*) Source: Authors' own elaboration from data [28–30].

Figure 4 shows that olive tree cultivation developed later and more slowly. It began in the lower regions of the mountain chain, and later spread up the mountainsides to almost 800 metres above sea level. The expansion of olive groves required capital to plant the olive trees and wait for their fruition some years later, as well as to construct the terraces to consolidate the plantations and the olive mills to press the olives and obtain oil. Commercial reasons were also behind, the olive groves taking a full century to find their expansion niche. Their full development as one of the main export crops of the island required the interconnection of agro-ecologically complementary markets: the Atlantic regions, where olive cultivation was not possible, and the Mediterranean. Population growth also meant a growing demand for olive oil for domestic consumption (food, lighting, food conservation) and manufacturing (wool, soap). This explains why the business of olive cultivation did not flourish until about the second half of the 15th century [116,117]. Olives and cereals were formerly cultivated on the grapevine terraces, whereas the sheep pastures contributed decisively to the spread of wild grasses such as the common reed grass, which was frequently burnt to renew the pasture. Wool and cheeses found an outlet in extra-regional expansion, favoured by royal privileges. The kingdom's protectionist policies fostered the development of a potent drape making business that not only covered internal demand, but also exported to the Levantine markets [110,111].

Therefore, right from the beginning the land concentration processes were linked with the expansion of urban textile manufacturing and the exportation of cloth to the Levantine markets, and later oil exports. The new estates required much land and capital (livestock, terraces and facilities), but they were less labour intensive than the farm units destined for cereal and viticulture. Only olive picking required an intense workforce mobilization. The implementation of these new production strategies, and the changes in how the large estates were managed, ushered in important changes in the organization and exploitation of labour.

## 6. The Emergence of a Low-Wage Labour Market in the 16th Century

Nonetheless, the transition to a wage labour force in Mallorca was not immediate. During the 14th century and the first half of the 15th the manorial estates and farms combined farm-workers with slaves to till their lands and drive their herds, together with both male and female wage labour hired by the day for harvesting, olive picking and reaping [50,73,95]. Some accounting books from the 15th century show the continuity of this model. In the period 1446–1460 the farm of Bellpuig, owned by the Vivot family of merchants, leased flocks of sheep and pastures. The cereal cropping was managed by a foreman, who organized the land cultivation using both free farm-workers and slaves. The hiring of day labourers was relatively scarce, and only took place during the months when the cereals were harvested [106] (pp. 83–108). The bookkeeping of Son Sureda farm during the decade of the 1530s shows a similar picture: the combined use of free farm-workers and slaves, supplemented by day labourers hired in the high seasons related to the cereal harvest and olive picking [95].

The labour regulations of 1351 and 1411 denote the presence of wage labourers on the manorial estates and farms. They set the wages and working conditions of the farm-workers that lived on the farms and were hired by the year, of the labourers hired by the day (1351), and they also addressed the issues of the olive pickers' negotiation capacity and their frequent unrest (1411). Nonetheless, most of these labour legislations were still aimed at controlling the slave workforce, abating towards the end of the 15th century and especially in the 16th. Significantly, the new by-laws set in 1550 and 1575 were designed to counter the wage demands of the new workforce hired on daily basis. They did not just fix the maximum wage for reaping, but they also regulated other tilling tasks like hoeing, digging and pruning the vines, and digging and weeding the cereals. The important role women played in weeding was mentioned for the first time in these labour regulations [73].

Table 7 summarises the changes in the use of labour between 1350 and 1600 in these three estates located in different regions of the island: Masnou in Alaró parish placed in Tramuntana region (1350–1356), at the foot of the mountain chain; Son Costa (1597–1602) in Montuïri village, at Pla region; and Son Gallard (1619–1623) in the district of the city of Palma. Cereals, vines and garden vegetables

were cultivated on all three estates. The percentages of each labour category and the gender of the free workforce hired, or of the slaves, are measured by the total number of days of work for each period and farm. The average annual numbers of work days requiring hired labour in the middle of the 14th century were 2500 in Masnou, 3800 in Son Costa, and 3500 in Son Gallard, respectively. These tree exceptional samples would show the transition from a typical slave farming shown in the probate inventories prior of 1450, and the new farms described in the inventories after 1500.

**Table 7.** Composition of free and slave labour used on three large farms to cultivate cereals and vines, in days of work and wages as percentages of the total on each estate.

| Masnou Farm, (Raiguer) 1350–1356 | % Slaves | % Labour Hired by the Day | % Foreman and Farm-workers | % Total |
|---|---|---|---|---|
| Women | | 3.2 | | 3.2 |
| Men | 49.8 | 18.1 | 28.9 | 96.8 |
| Total | 49.8 | 21.4 | 28.9 | 100.0 |
| **Son Costa Farms (Pla), 1597–1602** | **% Slaves** | **% Labour Hired by the Day** | **% Foreman and Farm-workers** | **% Total** |
| Women | | 33.9 | | 33.9 |
| Men | | 33.3 | 32.9 | 66.1 |
| Total | | 67.1 | 32.9 | 100.0 |
| **Son Gallard Farm (Palma) 1619–1623** | **% Slaves** | **% Labour Hired by the Day** | **% Foreman and Farm-workers** | **% Total** |
| Women | | 32.8 | | 32.8 |
| Men | | 25.9 | 41.3 | 67.2 |
| Total | | 58.6 | 41.3 | 100.0 |

Sources: Authors' own elaboration from [50,51,73].

As the Masnou farm account book and the other sources indicate, in the middle of the 14th century slaves made up the greatest part of the workforce, measured in the number of workdays required. The foremen and farm-workers played a supporting and supervising role with respect to the slave workforce, and the agricultural work in general. The male day labourers were basically hired to do some of the tasks needed to maintaining the vines (hoeing and pruning), while the female workforce was only hired for grape harvesting and olive picking. On the Masnou farm, male and female hired day labourers represented a fifth of the total workforce, a situation that was modified during the first half of the 15th century. The account books of the Son Costa and Son Gallard farms show how slavery had disappeared by the second half of the century. Day labourers represented around 60% of the total workforce, while the other 40% were basically farm-workers and the foreman, who were contracted by the year.

All the available sources assembled indicate that by the middle of the 15th century wage labour began to be more profitable than slavery. First, it dispensed of the need for institutional coercion procedures and slave work supervision. Second, it freed up a large part of the fixed investment in buying human beings as capital and covering their full maintenance (food, clothes and shelter) for other uses. Conversely, a large part of the economic expenses for reproduction of the free wage workforce was transferred to the peasant families where the women bore the cost of biological reproduction, and of the domestic care of children and the elderly [118]. Yet, all these factors did not prevent slave profitability while the low population densities after the Black Death and the access of the peasants to the open colonization frontier of Mallorca kept the wages of free labour at a high level. They only became decisive after the deep plebeian indebtedness due to the heavier royal taxes, the defeat of the peasant revolt of 1450–1454, and the widespread land grabbing that created a dispossessed peasantry and a low-wage labour market [33].

In the context of an increasing supply of low-wage labour hired to carry out a less labour-intensive productive specialisation, first in livestock and then in olive groves, the exploitation of a free wage workforce became more advantageous. An original and very characteristic feature of this new agricultural wage workforce was the growing participation of young rural women. Day work for hired labour during seasonal peaks became an extremely feminised labour market in Mallorca.

In the middle of the 15th century, 70% of day labourers and almost 50% of the total workforce were women. Meanwhile, the male workforce was becoming increasingly confined to the annual hiring as farm-workers. This labour segmentation by gender, age and time created a long-lasting organisational structure of the agricultural workforce of the large estates of the island. It was firmly established by the 17th century and would last until the 19th century [119]. However, the changes in labour market and the feminization of work-force did not change the gender wage gap, in the same vein as Humphries and Weisdorf [120,121] pointed out for the United Kingdom. In the 1350 the wage gap for the same tasks performed in Mallorca were less than a half, and between 1350 and 1580 it was kept as half. In 1575 an ordinance legally established that '*the female wage cannot be more than half of the male wage*' [122] (pp. 121,122) [123] (I p. 142, II, pp. 802,803). In Mallorca the massive incorporation of women in the farm labour market did not meant any change in the wage gender gap up to the 20th century [51]. This evidence reinforces the interpretation that it can only be understood through the prevalence of a patriarchal exploitation that had nothing to do with productivity differentials [120,121].

## 7. Concluding Remarks

From the very beginning slavery was present in the feudalisation process in Mallorca, accompanying its development until the 15th century. The slow, weak attraction of peasant settlers to colonize the island, coupled with the slow pace of population growth and the impossibility of imposing servitude on them contrary to what happened in northeastern 'Old Catalonia' where they first came from, led the feudal lords to capture and use slaves to toil the land. First, the descendants of the Andalus population that lived on the island were forced into slavery. Later on imported human beings from the eastern Mediterranean markets became the source of slave workers used in the Mallorca countryside. There is no evidence of any attempt to implement a system of demographic reproduction of that late feudal slavery. Control of the slave workforce was guaranteed by a comprehensive legal system, together with a private and public military system of vigilance and repression that prevented slaves from escaping or rebelling. The chance given to older slaves to buy freedom from their masters helped their amortisation as 'human capital', and also contributed to keeping this feudal slavery under control.

Paradoxically, the end of this late feudal-colonial slavery in Mallorca coincided with the beginning of the early modern slavery system deployed by the new mercantile capitalism in the New World. The European expansion of this new colonialism related slave plantations on the frontiers that produced sugar, coffee and tobacco with the emerging demands of new intermediate social groups who were becoming wealthier in the Atlantic Europe. The plantation enclaves of the 15th and 16th centuries were the laboratories of the large-scale colonial slavery later implanted in the Caribbean, South America, Africa and Asia. At the same time, however, on the island of Mallorca peasant dispossession and the land accumulation in the hands of a new, merged class of nobles and merchants paved the way for forming large estates, called *possessions*, at first devoted to land extensive and less labour-intensive livestock rearing.

Last, population growth and the restrictions imposed by that Mallorcan agrarian class structure on the peasants' access to land created a growing supply of low-wage labour from 1550 onwards. When this wage workforce became cheaper than slavery, a new pattern of agricultural labour management had to be organized on the island. The stable farm-worker on the *possessions* replaced the slaves in their habitual tasks: taking care of livestock grazing and cultivating the land. The seasonal peaks requiring labour to maintain soil fertility and weeding, and even more intensively for harvesting, posed an additional challenge. The large estates had to hire a huge number of day labourers who tried to take advantage of those work peaks which lasted just a short time to negotiate higher wages. The original, albeit little studied feature of this new labour regime was the incorporation of many more women in the agricultural workforce than in any other Mediterranean and Atlantic region at that time. This characteristic would differentiate the agricultural wage labour market of Mallorca from many others that developed in Europe in the 16th and 17th centuries.

**Supplementary Materials:** Supplementary materials can be found at http://www.mdpi.com/2071-1050/11/1/168/s1.

**Author Contributions:** The article conceptualization has been mainly carried out by G.J.-A. and E.T.; the methodology, by G.J.-A., A.M.-F. and R.S.-C.; empirical validation, by G.J.-A., A.M.-F., R.S.-C. and E.T.; sources compilation and investigation, by G.J.-A., A.M.-F. and R.S.-C.; the writing to the original draft preparation, by G.J.-A.; the writing-review & editing, by G.J.-A. and E.T.; and funding acquisition, by G.J.-A. and E.T.

**Funding:** This research was funded by Ministerio de Economía y Empresa del Gobierno de España grant numbers HAR2014-54891-P and HAR2015-69620-C2-1-P.

**Acknowledgments:** We are grateful for the comments and suggestions made by participants and attendees at Session 143 of the international conference Transrural (Santiago de Compostela 20–23 of June 2018), especially those made by Mathie Arnoux, José M. Lana Berasáin and Antoni Furió.

**Conflicts of Interest:** The authors declare no conflict of interest.

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
