# Peer review of "Socioecological Transition in Land and Labour Exploitation in Mallorca: From Slavery to a Low-Wage Workforce, 1229–1576"

_sustainability, doi:10.3390/su11010168_

Round 1

Reviewer 1 Report

This paper explains the change in slavery disappeared and the emergence of low-wage labor market in Mallorca Island where the largest slave market in the Mediterranean was located. In order to explain this change, this study is limited to representative aristocratic groups who own slaves and farmland in Mallorca Island, and shows basic statistical analysis reflecting the geographical and socioeconomic situation at that time.

This paper has a potential to be accepted, but some important points have to be clarified or fixed before we can proceed and a positive action can be taken.

We here summarize these points:

1.       Considering that it is a difficult situation to deduce the things at that time using limited data, the economic conditions shown in table 1 are very lack. Part 3, which explains feudal colonization and slave labor, explains the background of this period. In order to provide direct information on the economic situation and class at the time, it is necessary to present information that can be grasped in relation to many social and economic changes during the period. (Eg, the characteristics of the Mallorca region, the population and number of classes in Mallorca (age distribution), race and gender of slaves, labor productivity of slaves, etc.) That is, as long as the data permit, the characteristics of slaves and aristocrats that can inform the situation of the island of Mallorca at the time of major changes such as demographic decline and social conflicts should be presented.

2.    In the table and figure analysis presented by the authors, there is a disadvantage that the analysis year and year interval are arbitrary and the readability is poor. In particular, Tables 2 and 4 are extracted from the same sample of the same period, but the analysis period is different. Therefore, the reasons for such differences should be specifically addressed.

3.    Table 6 describes the different regions of the island that were not mentioned before. This table is not convincing because it does not specifically address the reasons for selecting three area that have not been mentioned before to show the change in labor demand. Also, in order to confirm the change, it is necessary to show both the previous value and the subsequent value for each region simultaneously. If this is not possible due to the limitations of the data, evidence should be provided that the three areas are similar areas with similar characteristics.

Reviewer 2 Report

Referring to above ratings - the topic of the paper is interesting but its lack of clear narrative makes it really difficult to follow.

It is not clear enough how the aim (to bring this debate to southern Europe placing it at the interface of socioeconomic and sociological conflicts) was realized. I cannot find what kind of socioeconomic and sociological conflicts are referred to.

The paper lacks a figure/graph which can guide through its structure and logic. A structure is described but then particular sections refer to mixed issues. It is difficult to go through the paper. A time line with main changes in all considered aspect should be added. I would also advise a review of particular sections in order to make it more separable (covering separated issues, no so mixed with each other).

Some details:

There are some typo mistakes, as line 326: “geograohical”, 489: “juan”?, 501: “wellas”, 536: “vaquer”

Legend for figure 1 – I cannot see a difference between current and historical wetlands

Line 505-506: “Figure 1 shows the percentual composition of the agriocultural production of Mallorca based on the tithe data” Is it true if figure no 1 is entitled “Map of the Mallorcan regions”?

It is not clear enough how “simultaneous changes in the relative prices of these products” can effect on conclusions about evolution of production? 498-504

I cannot find a reference to figure 4 in the text.

Table 6 – value on the cross of men and %total 78.6% does not seem appropriate.

Reviewer 3 Report

- The authors discuss about gender segregation, gender pay gap, concentration of women in low pay occupation in an historical perspective. It could interesting a link with present situation. Discrimination and gender pay gap are also actually evident for apical position/managerial which are the other extreme of the wage distribution with respect to the low wage position in the paper. Social attitudes, gender stereotypes and cultural biases may constitute the most relevant obstacle to equality between men and women in terms of wages. it is evidente also in Spain: see...Garcıa et al. (2001) Scicchitano (2014a, IJM), Gardeazabal and Ugidos (2005). Other relevant papers for other countries   are Bertrand and Hallock (2001), Muñoz-Bullon (2010), Biagetti and   Scicchitano (2011), Bertrand et al. (2010) Yurtogly and Zulehener (2009),   Smith et al. (2011), Scicchitano (2011, 2014b). This literature should be added in other to explain that actual gender discrimination derives from a number of centuries 

- Total in first row of table 6 should be checked
